# Association between Motivational Climate, Adherence to Mediterranean Diet, and Levels of Physical Activity in Physical Education Students

**DOI:** 10.3390/bs9040037

**Published:** 2019-04-11

**Authors:** Gabriel González-Valero, José Luis Ubago-Jiménez, Irwin A. Ramírez-Granizo, Pilar Puertas-Molero

**Affiliations:** Department of Didactics of Musical, Plastic and Corporal Expression, University of Granada, 18071 Granada, Spain; ggvalero@ugr.es (G.G.-V.); jlubago@ugr.es (J.L.U.-J.); irwin@correo.ugr.es (I.A.R.-G.)

**Keywords:** motivational climate, Mediterranean Diet, physical activity, students, Physical Education

## Abstract

Physical Education is an essential educational area to develop physical-healthy habits and motivational orientations, which are fundamental to guide the situation of future Physical Education teachers. These professionals will have a fundamental role in teaching different types of motivations, active lifestyles, and healthy habits in youths. For this reason, the objective of the study is to know the association between motivational climate, adherence to the Mediterranean Diet (MD), and the practice of physical activity in future Physical Education teachers. A cross-sectional and nonexperimental study was carried out using a single measurement within a single group. The sample consisted of 775 university students from the cities of Andalusia (Spain). Motivational climate was evaluated through the Perceived Motivational Climate in Sport Questionnaire (PMCSQ-2), levels of physical activity were evaluated through the adolescent version of the Physical Activity Questionnaire (PAQ-A), and level of adherence to the MD was assessed through Mediterranean Diet Quality Index (KIDMED). On one hand, the healthy and self-improvement component promoted by physical activity favors an orientation focused on process and learning. Likewise, the competitive component is key to motivation focused on product and social recognition. In addition, future Physical Education teachers should pay special attention to the unequal recognition among members that physical activity can generate, in order to avoid personal disregard and social rejection. The ego climate is related to a high adherence to the MD. On the other hand, the future Physical Education teachers who manifest motivational processes based on fun and their own satisfaction have low levels of adherence to the MD.

## 1. Introduction

Numerous studies have found a positive relationship between the performance of physical sporting activities and psychosocial, physical, and health benefits. These benefits improved the quality of life, emotional well-being, and motivation towards this practice in young adolescents. This life stage represents a period in which behavior patterns are consolidated and easily influenced [1]. For this reason, it is vitally important to work on motivation in order to establish healthy habits to the detriment of disruptive behaviors that will be the basis of adult behavior [2,3,4].

Ongoing research highlights that the development of motivational aspects within the classroom determines a greater persistence of physical sports practice and the acquisition of healthy habits inside and outside the classroom [5,6]. The motivational theories that have facilitated the understanding of motivation in the physical sports and educational context in recent decades have been the theory of achievement goals [7,8] and the theory of self-determination [9,10]. The theory of achievement goals postulates that a student’s involvement towards a task or ego orientation originates from the confrontation between the motivational orientation of young people and the motivational climate they perceive [8]. This is why the teacher is a fundamental element generating one type of climate or another [11,12]. 

This theory of task orientation or mastery is characterized by the use of self-references, as it does not care about the achievements of others, but their own progress. In fact, they are considered to be competent and successful if, every day, one improves in relation to oneself [13,14]. It is also associated with intrinsic motivations achieved through fun and satisfaction, cooperation, commitment, participation, and continuity in practice [15,16]. However, ego or performance orientation is associated with the use of references to evaluate success and competition in comparison to other persons [17,18]. At the same time, this type of goal is associated with extrinsic motivations achieved thanks to the recognition and social status and the physical sports and educational failure of others [19,20]. Most of the recent research has studied the motivational climate of sport and has focused its objectives on sports performance and the predisposition to follow a healthy and active lifestyle [21,22].

Maintaining a healthy lifestyle during development improves physical, psychological, and social fitness. For this reason, motives related to health and well-being, such as a healthy diet and sports practice, are maintained throughout life as motivational factors in the sporting and educational context [23,24]. Eating habits are essential to achieve physical and intellectual development, as well as to prevent cardiovascular health problems resulting from the impoverishment of the diet [25]. In fact, Cervera, Serrano, Vico, Milla, and García (2013) [26] highlight that the diet followed by the university population is based on malnutrition, suppression of essential foods, and the intake of refined sugars and foods high in fat [27]. It is at this point that adherence to the Mediterranean Diet (MD) comes into play, as multiple studies show that MD is related to health benefits, such as increased life expectancy, adherence to physical activity, and decreased cardiovascular and psychological diseases [28,29]. MD is characterized by the consumption of foods originating in Mediterranean lands, highlighting olive oil, fruits, vegetables, cereals, and beans, in addition to balanced egg, dairy products, and fish consumption [30].

Physical Education plays an essential role in the development of healthy physical habits, emphasizing that the motivation provided will be essential to guide the current and future situation of Physical Education teachers. These professionals will be decisive in the configuration of motivational climates, active lifestyles, and the acquisition of healthy habits by young people at an early age. For this reason, the objective of the study is to know the association between motivational climate, adherence to the MD, and the practice of physical activity in future Physical Education teachers.

## 2. Materials and Methods

### 2.1. Subjects and Design

A cross-sectional and nonexperimental study was carried out using a single measurement within a single group. The sample consisted of 775 university students from the eight Andalusian cities (Spain), with a gender representation of 43.1% (n = 320) female and 58.7% (n = 455) male. The age range of the participants was between 20 and 29 years old (22.22 ± 3.76). Selection criteria of the sample required all participants to be studying the Degree in Primary Education with a specialist in Physical Education in Andalusia. According to the data provided by the administrations of the different universities, the students enrolled in courses including a Physical Education specialist in Andalusia, totaled 1167 for the 2016/2017 academic year.

### 2.2. Measures

Motivational climate was evaluated through the Perceived Motivational Climate in Sport Questionnaire (PMCSQ-2), adapted to Spanish by González-Cutre, Sicilia, and Moreno (2008) [31] and originally developed by Newton, Duda, and Yin (2000) [32]. This questionnaire is comprised of two dimensions which pertain to the subscales Task-Climate (TC) [Cooperative Learning (CL), Effort/Improvement (EI) and Important Role (IR)] and Ego-climate (EC) [Punishment for Mistakes (PM), Unequal Recognition (UR), and Rivalry between Membership (RM)]. This scale is composed of 33 items rated on a five-point scale (1 = Strongly disagree; 5 = Strongly agree). Examination of internal consistency produced an acceptable value for Cronbach’s alpha (α = 0.82).

Levels of physical activity were evaluated through the adolescent version of the Physical Activity Questionnaire (PAQ-A) [33] translated to Spanish by Martínez-Gómez et al. (2009) [34]. This instrument produced an overall score by summing the ten items, of which each is scored on a five-point Likert scale where 0 is “Never” and 4 is “Always”. This measure is used to evaluate the level of engagement in physical activity during the week prior to measure completion. An acceptable Cronbach alpha was obtained in the present study (α = 0.80).

Level of adherence to the MD was assessed through Mediterranean Diet Quality Index (KIDMED) [35]. This scale is composed of 16 dichotomous items which can be answered as yes or no. There are 12 positively-framed and four negatively-framed items. In the case of these items, a score is obtained which ranges from −4 to 12. For this scale, acceptable internal consistency was identified highlighted, with a Cronbach alpha of α = 0.83.

### 2.3. Procedure

Firstly, collaboration of the universities and participants was requested through information packs developed by the department of Didactics of Musical, Plastic and Bodily Expression of the University of Granada. Packs were administered to university students enrolled on a Primary Education teaching course with a specialty in Physical Education, in one the eight Andalusian cities. Packs were administered through the different departments of the universities and provided information about the nature and objectives of this study. In addition, informed consent was requested to participate in the study. Secondly, students were informed about the data collection process. The tests and scales were applied during university teaching hours without any incentives being offered. In addition, researchers were present to help participants with possible difficulties and to ensure anonymity of the data. Further, the Ethics Committee of the University of Granada approved the study (462/CEIH/2017) and ethical principles established by the Declaration of Helsinki for research were followed.

### 2.4. Statistical Analysis

Descriptive analysis was carried out using the software SPSS® version 22.0 (IBM Corp., Armonk, NY, USA). Normality of the data was tested using the Kolmogorov–Smirnov and Shapiro–Wilk tests, and it was observed that the values followed a normal trend, so parametric tests were used. For theses analyses, the Pearson Chi-square test was used and the level of significance was set at 0.05. To see the differences between the variables, single-factor ANOVA was used. Pearson's bivariate correlations were used at the significance level of *p* < 0.05 and *p* < 0.01.

## 3. Results

Table 1 shows the motivational climate according to the level of physical activity. High levels of physical activity were significantly related (*p* < 0.05) to orientation towards the EC (M = 2.27; S.D. = 0.809) and TC (M = 4.14; S.D. = 0.587), with higher levels in the latter. Similarly, the highest mean values of physical activity were significantly associated (*p* < 0.05) with CL (M = 4.21; S.D. = 0.665), EI (M = 4.08; S.D. = 0.587), PM (M = 2.12; S.D. = 0.855), and RM (M = 2.83; S.D. = 1.019). However, it was demonstrated that students with low levels of physical activity were significantly related to UR (M = 2.17; S.D. = 0.982) (*p* = 0.025).

Table 2 shows the motivational climate as a function of the level of adherence to MD. Low adherence to MD is significantly related (*p* < 0.05) to TC (M = 4.48; S.D. = 0.323), CL (M = 4.60; S.D. = 0.318) and EI (M = 4.33; S.D. = 0.443). On the other hand, adherence to high MD is significantly associated (*p* < 0.05) with orientation towards EC (M = 2.16; S.D. = 0.790), PM (M = 2.13; S.D. = 0.853) and RM (M = 2.64; S.D. = 0.994).

Table 3 shows the correlation coefficients between motivational climate with physical activity levels and MD. Climate was associated with levels of physical activity and MD at the level of significance of *p* < 0.05 and *p* < 0.01. Physical activity levels were positively associated with TC (r = 0.235), CL (r = 0.224), EI (r = 0.242), IR (r = 0.193), EC (r = 0.210), UR (r = 0.196), and RM (r = 0.244). Adherence to MD is positively associated with RM (r = 0.258), EC (r = 0.198), UR (r = 0.183) and RM (r = 0.230).

## 4. Discussion

In the present research, in which 775 students of Physical Education participated, the association between motivational climate, adherence to the MD, and physical activity practice was investigated. The peculiarity of this study lies in studying these associations in a specific population; future Physical Education teachers. One of the learning methodologies innate in young students is the observation and imitation of teachers, which explains the importance of researching this population, since it will be their attitudes and habits that children learn and reproduce in the future. However, with a similar nature, studies are highlighted in which these variables are described and related in the population of students and sportsmen [36,37].

The practice of vigorous physical activity is clearly related to the orientation towards the EC. The competitive component implicit in this practice is a key factor in this type of performance orientation. Likewise, physical activity is related to the body’s well-being [38], which is linked to satisfaction and triumph thanks to social recognition. There is a reality where people who turn to the ego experience high levels of pressure to reach a goal, causing the abandonment of sports practice [39,40,41].

Similarly, task orientation is also linked to high levels of physical activity. There are several studies that emphasize that the participants with a greater orientation to the task acquire a great adherence to the physical sports practice [42] due to the fact that they carry it out for satisfaction or recreation. This contributes positively and generates persistence in the actions carried out [43]. Physical activity is an effective means for personal well-being; therefore, this practice favors a motivation that focuses on self-improvement, personal benefit, and continuous learning.

Even so, in the academic and sports fields, special attention must be paid to the unequal recognition among members, since due to the competitiveness implicit in games and sports, situations may arise in which personal disregards appear, and even social rejection may appear [44]. Research shows that in interventions created to increase motivation and create physically healthy habits in young people, strategies should be considered that favor the perception of competencies and skills of young people, while alternating motivational climates oriented to the task and the ego [45,46].

Over time, it has been shown that adequate nutrition is associated with increased sporting performance [47,48]. Based on these premises, the orientation towards the ego climate is associated with a high adherence to the MD. In fact, dietary care is an essential factor in competition with adversaries [49]. On the other hand, subjects who tend to orient themselves towards motivational levels focused on processes and personal well-being (task climate) are associated with low levels of adherence to the MD [22].

The main reason for this association is that people who care about having fun and their own satisfaction do not give importance to dietary care. Even so, it is highlighted that effort and personal improvement are directly associated with adherence to the MD [50]. Studies such as Balaguer, Duda, and Castillo (2017) [51] and Moreno-Murcia et al. (2015) [41], show that task orientation is positively related to physical and healthy practice, while it is negatively related to harmful substance consumption.

## 5. Conclusions

Motivational orientations towards tasks and the ego are associated with physical activity. On one hand, the health and self-improvement component promoted by physical activity favors the learning process. Likewise, the competitive component is key in motivation centered on the product and social recognition. In addition, future Physical Education teachers should pay special attention to the unequal recognition among members that physical activity can generate, in order to avoid personal disregard and social rejection.

The EC is related to a high adherence to the MD. The reason for this is that food is a key factor in performance and rivalry between members. On the other hand, the future Physical Education teachers who manifest motivational processes based on fun and their own satisfaction have low levels of adherence to the MD. However, it should be noted that effort and personal improvement are directly associated with adherence to the MD.

The main limitations of the study are that it is a cross-sectional study, which means that cause–effect assessments cannot be established. Other variables could also be included that work on attitudes towards Physical Education, aspects related to mental health, and the teaching–learning process. Likewise, the main virtue of this research may also be a limitation, since it has been carried out on a specific population of university students of Physical Education. This importance is also highlighted when it comes to student learning, since this will reproduce the habits and behaviors of teachers. Finally, as future perspectives for the study, the aim is to work with variables related to psychological well-being and to propose interventions in which motivational orientations centered on process and performance are combined with physical sports and mental activity programs.

## Figures and Tables

**Table 1 behavsci-09-00037-t001:** Motivational climate and physical activity level.

Motivational Climate	PAQ-A	Media	S.D.	F	*X* ^2^
**TC**	**Low**	3.89	0.640	5.004	0.007 *
	**Medium**	3.98	0.631		
	**High**	4.14	0.587		
**CL**	**Low**	3.95	0.751	6.231	0.002 *
	**Medium**	3.99	0.740		
	**High**	4.21	0.665		
**EI**	**Low**	3.82	0.625	5.146	0.006 *
	**Medium**	3.92	0.649		
	**High**	4.08	0.587		
**IR**	**Low**	3.96	0.805	1.966	0.141
	**Medium**	4.07	0.748		
	**High**	4.17	0.689		
**EC**	**Low**	2.12	0.755	5.811	0.003 *
	**Medium**	2.04	0.730		
	**High**	2.27	0.809		
**PM**	**Low**	1.94	0.840	3.770	0.023 *
	**Medium**	1.92	0.785		
	**High**	2.12	0.855		
**UR**	**Low**	2.17	0.982	3.700	0.025 *
	**Medium**	1.96	0.858		
	**High**	2.10	0.955		
**RM**	**Low**	2.51	0.822	8.220	0.000 *
	**Medium**	2.49	0.920		
	**High**	2.83	0.019		

TC: Task Climate; CL: Cooperative Learning; EI: Effort/Improvement; IR: Important Role; EC: Ego Climate; PM: Punishment for Mistakes; UR: Unequal recognition; RM: Rivalry between membership. *p* < 0.05 (*).

**Table 2 behavsci-09-00037-t002:** Motivational climate and MD adherence.

MD Adherence		Media	S.D.	F	*X* ^2^
**TC**	**Low**	4.48	0.323	4.529	0.011 *
	**Medium**	3.95	0.624		
	**High**	4.05	0.626		
**CL**	**Low**	4.60	0.318	3.855	0.022 *
	**Medium**	3.98	0.734		
	**High**	4.07	0.727		
**EI**	**Low**	4.33	0.443	6.933	0.001 *
	**Medium**	3.87	0.650		
	**High**	4.02	0.619		
**IR**	**Low**	4.62	0.354	1.956	0.142
	**Medium**	4.07	0.724		
	**High**	4.08	0.761		
**EC**	**Low**	1.81	0.703	2.955	0.026 *
	**Medium**	2.06	0.715		
	**High**	2.16	0.790		
**PM**	**Low**	1.66	0.372	3.654	0.041 *
	**Medium**	1.91	0.759		
	**High**	2.13	0.853		
**UR**	**Low**	1.77	0.971	1.280	0.256
	**Medium**	2.01	0.865		
	**High**	2.02	0.916		
**RM**	**Low**	2.19	1.033	3.031	0.049 *
	**Medium**	2.49	0.885		
	**High**	2.64	0.994		

TC: Task Climate; CL: Cooperative Learning; EI: Effort/Improvement; IR: Important Role; EC: Ego Climate; PM: Punishment for Mistakes; UR: Unequal recognition; RM: Rivalry between membership. *p* < 0.05 (*).

**Table 3 behavsci-09-00037-t003:** Correlation coefficients between motivational climate, physical activity level, and MD.

	Motivational Climate and dimensions							
	TC	CL	EI	IR	EC	PM	UR	RM
**PAQ-A**	0.235 **	0.224 **	0.242 **	0.193 **	0.210 **	0.065	0.196 **	0.244 **
**KIDMED**	0.011	−0.012	0.258 **	−0.024	0.198 **	0.183 *	0.066	0.230 **

TC: Task Climate; CL: Cooperative Learning; EI: Effort/Improvement; IR: Important Role; EC: Ego Climate; PM: Punishment for Mistakes; UR: Unequal recognition; RM: Rivalry between membership; PAQ-A: test aggregate; KIDMED: test aggregate. *p* < 0.01 (**); *p* < 0.05 (*)

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
