# Peer review of "Association between Motivational Climate, Adherence to Mediterranean Diet, and Levels of Physical Activity in Physical Education Students"

_behavsci, 2019, doi:10.3390/bs9040037_

Round 1
Reviewer 1 Report
There are a variety of English corrections needed. I have copied and pasted the pdf file into a Word file (without taking out the line numbers). I have made a set of suggested corrections which you should check in case I have changed your meaning. Otherwise this is a well conducted study, clearly expressed and of general interest. With changes made to the language and in the text just before Table 1 you need to spell out the acronymns EC TC CL etc in the text the first time you use them.

Author Response
For both reviewers a style review has been performed throughout the manuscript.
Reviewer 1
En primer lugar agradecemos las sugerencias aportadas. En la línea 73 se ha sustituido el término “legumes” por “beans”. Atendiendo a otra de las sugerencias, en la línea 88 y 89 se ha especificado que la muestra se corresponde a toda Andalucía, suprimiendo la expresión anteriormente expuesta. En la línea 118 se ha especificado “… of this study”.
En referencia a la modificación de los acrónimos, se han especificado las siglas en el apartado “measures” además de aparecer en el pie de página

Reviewer 2 Report
Thank you for the opportunity to review this manuscript.
I recommend improving your abstract because lines 12-18 are duplicate with line 75-81.
In last decade your topic is well studied and in this regards I recommend highlighting the novelty of your study.
I recommend revising the Discussion to be more specific to your finding.
I recommend adding the limits and strength points of your study.
The Conclusions are too generally and I recommend you focusing only on specific conclusions.
Author Response
For both reviewers a style review has been performed throughout the manuscript.
Reviewer 2
In relation to the first comment suggested for the abstract, it has been modified to make sense of it and avoid it coinciding with other parts of the manuscript.
In the first paragraph of the discussion, the novelty of this study has been highlighted. On the one hand, it is stated that the population of Physical Education teachers in training is not very well studied. In addition, since students learn by observation and imitation, it will be necessary to know what these characteristics of the teachers will be, since in this way they will be shown to the students and they will reproduce them.
Based on the recommendation of the specificity of the discussion, we start from the bibliographic limitation in terms of studies that work on the variables proposed in the research and in physical education teachers in formation. For this reason, we have had to rely on studies that worked with athletes or students.
If the reviewer considers it appropriate, as we have decided, the study limitations, strengths and future prospects have been added in the last paragraph of the conclusions.
Finally, and based on the fact that the commentary states that it recommends that we make more specific conclusions, we believe that the first two sentences of the first two paragraphs of the conclusions are expressed in the manner requested by the reviewer. However, in order to give meaning and coherence to the study (since we have previously been asked to stress the importance of the study), it seems appropriate to keep those small comments that can be related to the objectives and novelty of the study, so that possible readers who wish to replicate this study can intervene in the classroom and make a comparison between the teacher-student collective.

Round 2
Reviewer 2 Report
The authors improved the manuscript according with the recommendations.